# Monitoring the Risk of the Electric Component Imposed on a Pilot During Light Aircraft Operations in a High-Frequency Electromagnetic Field [note 1]

**DOI:** 10.3390/s19245537

**Published:** 2019-12-14

**Authors:** Joanna Michałowska, Arkadiusz Tofil, Jerzy Józwik, Jarosław Pytka, Stanisław Legutko, Zbigniew Siemiątkowski, Andrzej Łukaszewicz

**Affiliations:** 1The Institute of Technical Sciences and Aviation, The State School of Higher Education in Chelm, Pocztowa 54, 22-100 Chełm, Poland; 2Faculty of Mechanical Engineering, Lublin University of Technology, 20-618 Lublin, Poland; atofil@pwsz.chelm.pl (A.T.); j.jozwik@pollub.pl (J.J.); j.pytka@pollub.pl (J.P.); 3Institute of Mechanical Technology, Poznan University of Technology, 60-965 Poznań, Poland; stanislaw.legutko@put.poznan.pl; 4Faculty of Mechanical Engineering, Kazimierz Pulaski University of Technology and Humanities, 26-600 Radom, Poland; z.siemiatkowski@uthrad.pl; 5Faculty of Mechanical Engineering, Bialystok University of Technology, 15-351 Bialystok, Poland; a.lukaszewicz@pb.edu.pl

**Keywords:** electromagnetic fields (EMFs), dosimeter, high frequencies, aircraft operation, environment

## Abstract

High-frequency electromagnetic fields can have a negative effect on both the human body and electronic devices. The devices and systems utilized in radio communications constitute the most numerous sources of electromagnetic fields. The following research investigates values of the electric component of electromagnetic field intensification determined with the ESM 140 dosimeter during the flights of four aircrafts—Cessna C152, Cessna C172, Aero AT3 R100, and Robinson R44 Raven helicopter—from the airport in Depultycze Krolewskie near Chelm, Poland. The point of reference for the obtained results were the normative limits of the electromagnetic field that can affect a pilot in the course of a flight. The maximum value registered by the dosimeter was E = 3.307 V/m for GSM 1800 frequencies.

## 1. Introduction

The effect of electromagnetic fields on human health and various areas of human activity is studied in numerous scientific and research centers worldwide. Despite researchers’ widespread interest, the methodology for the assessment of the effect of electromagnetic fields has not been standardized. Considerable activities aimed at decreasing the negative influence of electromagnetic fields are being undertaken [1,2,3]. It is worth mentioning that there are plentiful positive applications of electromagnetic fields, particularly in medicine, such as its therapeutic effect in the treatment of such disorders as neoplasms, burns, circulatory system diseases, or arthritis, as well as modern diagnostic imaging methods that utilize electromagnetic fields, including computed tomography, optical tomography, and microwave tomography. Moreover, research on the effect of electromagnetic fields on human health, which is concerned with the operation of electric and electronic devices, are being conducted [3,4,5]. Except for the direct protection of health, there are numerous technical aspects regarding the hazards resulting from electromagnetic fields exceeding their normal values. High-frequency electromagnetic fields can disrupt or even cause permanent damage to electronic equipment used in aviation, among other areas. Therefore, some special attention is paid to the issue of electromagnetic fields that can disturb the operation of communication, radar, and location devices or systems. The research investigates avionic measuring equipment that is subjected to high risk during its operation [6,7,8,9,10].

Research on electronic devices used in aviation is performed in specially screened chambers where avionic equipment is subjected to high-frequency electromagnetic fields from 10 kHz to 40 GHz. The research is conducted in order to determine the sensitivity of the devices to the effects of external fields and whether the instruments affect the environment. The requirements, which must be met by all onboard electronic devices and systems used in aviation, are depicted in the EU Directive 2013/35/ [11,12,13,14,15,16]. Onboard devices, radio-navigation devices, aviation communication, and navigation systems constitute basic avionic equipment.

High-frequency electromagnetic fields are most frequently produced by radio communication devices. Radio stations, telecommunication devices, mobile telephony base stations, WiFi devices, radars, and communication systems are indispensable nowadays. Tests on the effect of electromagnetic fields on electronic devices are commonly conducted when designing the device [17,18,19].

The increasing load of the electromagnetic field in light aircrafts can lead to negative effects on the pilots’ health and mental condition, especially when it comes to instructors flying many-hour flights daily, which can lead to safety risks. Studies on the impact of electromagnetic fields on humans result from the EU Directive 2013/35/ directive. The continuous increase in radio infrastructure, including mobile telephony, is associated with an increasing electromagnetic field strength. In particular, in the area of Global System for Mobile Communications (GSM) and Universal Mobile Telecommunications System (UMTS) relays, the electromagnetic field values can reach high values. The second source of electromagnetic fields that affects the pilot during the flight is avionics. Integrated avionics, which use one large display instead of many single indicators, is a great convenience for the pilot due to the presentation of a great deal of flight information on one display. Glasscocpit is currently the standard in aircraft communication. Thus, flight training organizations (FTO) are increasingly willing to use training aircrafts with the Glasscocpit system [20,21,22,23,24].

The purpose of the present study was to conduct electromagnetic field measurements on selected aircrafts from the Aviation Training Center in Royal Depultycze, near Chelm, Poland. The center is an integral part of the higher vocational school in Chelm and conducts training for airline pilots on airplanes and helicopters as a part of engineering studies.

The results were depicted in the form of graphs as a function of time. Due to the stochastic character of the sample, the analysis of the data obtained, with the emerging characteristic trends and parameters, was performed using the Statistica 13. 3 software (JPZ803D036327AR-2, StatSoft Poland, Cracow, Poland).

## 2. Materials and Methods

### 2.1. Measuring Method

The method used during the research involved broadband measurements using broadband meters. The meters utilized for environmental measurements of an electromagnetic field (EMF) in the vicinity of the objects researched and the far field are commonly used in protective measurements. Such measurements were used in order to obtain a single result that corresponds to the field intensity values of all sources within the measuring range of the probe. 

### 2.2. The Device Used for the Measurements 

The Maschek ESM 140 (Serial No: 30171, Maschek Electronic, Bad Wörishofen, Germany) dosimeter was used for the flight test measurements. The device measures the electromagnetic field for broadband high frequencies in real time and the measured data was stored in the device’s memory. We transferred the data to a personal computer for analysis after each flight. The measuring range was 0.01 to 70 V/m, with a sensitivity of 10 mV/m and an accuracy of ±2 dB in a free field and ±4 dB when the device was installed on the pilot’s arm for in–flight measurements (see Figure 1, right) [1,3,4].

The ESM 140 dosimeter with a battery weighs 87 g, which is relatively light for a professional measuring device. As a result, the device is almost imperceptible for the pilot during many-hour flights. The ESM 140 m were applied in tests on several different flight routes performed by four types of aircrafts.

The measuring process was initiated the moment the pilot left the service room. The entire work space was subjected to the measurement, including all areas where the electromagnetic fields could be present. Such a method ensured full reproduction of the exposure to which the pilot was subjected to.

The aircrafts used for the tests were fixed on a rotary wing, as shown in Figure 2. Three fixed-wing aircrafts were: the Cessna 152, the Cessna 172, and the AT3. The rotary-wing aircraft used in this study was the Robinson R44 Raven. All the aircrafts were used in every day flight training in the Center of Aviation of The State School of Higher Education in Chełm, East Poland. 

All the aircrafts were of typical monocoque construction, made of duralumin. They had a piston engine drive and a set of typical on-board instruments for Visual Flight Rules (VFR) or Instrument Flight Rules (IFR) flights [25,26]. Selected technical data of the aircraft used in the test flights are collected in Table 1.

The Cessna 152 and Cessna 172 aircrafts had Glasscocpit avionics, with the Cessna 152 aircraft having been installed as part of the retrofitting of an existing panel. The Aero AT3 aircraft was the lightest among those tested and also had Glasscocpit avionics. The Robinson R44 Raven helicopter featured the classic Helicopter In Flight Refuelling (H-IFR) flight instrument set together with the Aspen Avionics EFIS (see Figure 2).

## 3. Results and Discussion

The measurements were performed during the flights. In order to measure the intensity of the electric field, the first measurement was conducted on the ground in the Air Traffic Control Tower of Depułtycze Królewskie airfield (Figure 3). 

Maximum values of the electric component of the electromagnetic field obtained during measurements in the ground service station were measured using the GSM 900 frequency band. What is of great significance is that the values were several-fold higher than the ones recorded in other frequency bands. The intensity of the electromagnetic field in this location was relatively low. 

The next step involved the measurement during the flights. The flights presented in the study were performed in the vicinity of such towns as: Krasnystaw, Bilgoraj, and Tomaszow Lubelski where antennas of mobile telephony are located. The selected flight route is shown in Figure 4.

The results for the GSM 900 system are shown in Figure 5. The values depicted in Figure 5, Figure 6 and Figure 7 refer to the measurement results obtained using the dosimeter in the Cessna 152. 

The highest measured value using the GSM 900 frequency band obtained was E = 0.29 V/m for the oldest mobile telephony system. GSM 900 frequency was the first system utilized in mobile telephony communication. The next GSM 1800 frequency band obtained in the measurement with the ESM 140 dosimeter is shown in Figure 6.

The maximum result of the intensity of the electric component of the electromagnetic field using the GSM 1800 obtained during the measurement with the dosimeter was E = 1.58 V/m.

Communication systems, such as UMTS, used for frequencies of 1920–2170 MHz, and GSM differ in terms of the implementation of various modern multimedia services. Moreover, UMTS uses services that are available both on the ground and through the satellite system. The system enables the simultaneous transmission of audio, video, and data in real time.

The maximum value obtained during the Cessna C152 flight for the UMTS communication system was E = 1.21 V/m (Figure 7). It can be observed that the maximum values of the electric field for all frequency bands analyzed refer to the point of time from the 68th to the 70th minute of the flight, which corresponds to the location of the aircraft over Bilgoraj. 

The measurement error was recorded for the frequencies researched during a flight, which are presented in Figure 8.

The limits of measurement uncertainty for the data obtained are given in Figure 8. Systematic error was chosen as the basis for uncertainty, whose sources were mainly: instrument class, characteristics of the band filters in the tested frequencies, flight altitude and heading relative to electromagnetic field sources, and meteorological factors. Moreover, the measurement error was also affected by the installation of the measuring unit, which, mainly for practical reasons, was placed on the pilot’s arm (see Figure 1, right), which meant that the measurement was not carried out in the free field. In future test, we are planning to use an on-board electromagnetic field-monitoring sensor in order to minimize the installation effects. This sensor is described in Section 4 of the present paper.

Although the majority of high-frequency electromagnetic field measurements are performed in scientific centers on land, this study was concerned with the comparison of the results obtained both before take-off and during the flight. The results confirmed that the pilot was more exposed to the effects of the high-frequency electromagnetic field during the flight.

The analysis presented next was for a two–hour training flight by the Cessna C172 on the following route: Depułtycze Królewske–Lublin Airport–Krasnystaw–Depułtycze Królewski. During the flight, the approach to landing on runway 25 at Lublin Airport was performed. Measurements of electromagnetic field for individual frequency ranges are shown in Figure 9, Figure 10 and Figure 11.

The highest value of the electric component of the electromagnetic field was recorded using the UMTS frequency band with E = 2.30 V/m. At this point, the aircraft had entered the instrument landing system (ILS) approach path. It is a radio navigation system that supports the landing of an aircraft in conditions of limited visibility. For the other frequencies tested, the electric field values obtained that affect the pilot was exposed are much lower. For the GSM 900 frequency band, the electric component of the electromagnetic field was E = 0.6 V/m, and for the GSM 1800, it was E = 1.05 V/m.

The next aircraft analyzed was the AT3. The flight took place on the route Depułtycze Królewskie–Lublin–Mielec–Depułtycze Królewskie (Figure 12, Figure 13 and Figure 14). The total flight duration was 1.40 h.

For the training flight with the AERO AT3 aircraft, the highest value was also recorded using the UMTS frequency band with E = 1.15 V/m (Figure 14). A similar value was recorded for the GSM 900 frequency with E = 1.14 V/m (Figure 13) and the lowest was for the GSM 1800 with E = 0.68 V/m.

Another analysis was carried out for the Robinson R44 helicopter flight (Figure 15, Figure 16 and Figure 17).

For the Robinson R44 helicopter flight, the highest value of the electric component of the electromagnetic field E = 1.89 V/m was read using the UMTS frequency. For the GSM 1800 frequency band, the maximum value of the electrical component was E = 1.16 V/m. For the oldest GSM 900 communication system, the maximum values (electric field) were two–fold lower than values obtained from measurements for other frequencies and equaled E = 0.89 V/m.

The statistical analysis involved the introduction of the values of the electric field intensity recorded by the ESM 140 dosimeter into the Statistica 13.0 software. The values of the analyzed parameters measured in the normal scale were characterized by the number and percentage, while those measured on a ratio scale by means of average, medians, standard deviation (SD), and range of variation. A 5% error of inference and associated significance level of *p* < 0.05 were adopted.

The data used in the statistical analysis were obtained during flights with all four aircrafts. First, results from a flight with the Cessna 152 aircraft were analyzed. The flight took 1.42 h on the route 1 from Depultycze Krolewskie to Bilgoraj and Tomaszow Lubelski using all frequency bands, from UMTS installations on the ground, as well as from onboard instruments (see Table 2). The following table presents the characteristics of the electric field intensity E for individual frequency bands of the selected route. A sample group included a total of 15641 measurements registered for each frequency band.

The mean value of the electric field intensity using the individual GSM and UMTS frequency bands ranged from 0.007 V/m to 0.048 V/m. The range of the variable analyzed was from 0.000 V/m to 3.30 V/m. Differences for the indicated frequency bands obtained with the ESM 140 dosimeter proved to be statistically significant and are depicted in Figure 18.

Selected results for one GSM1800 frequency range in which the highest values of the intensity of the electric component of the electromagnetic field were recorded during flights with one type of Cessna 152 aircraft was made on four routes. The variation of the electric field intensity for various aircraft routes is shown in Figure 19.

The data presented in Figure 19 consider the highest values of the electric field recorded for four aircraft routes for the GSM 1800up frequency band, which justifies the choice of the frequency band.

The numbers from one to four represent the flight routes researched, namely route 1: Depultycze Krolewskie-Bilgoraj-Tomaszow Lubelski, route 2: Depultycze Krolewskie–Rejowiec–Siennica–Pokrowka, route 3: Depultycze Krolewskie–Cycow–Rejowiec, and route 4: Depultycze Krolewski–Krasnystaw–Frampol. 

Variation in the intensity of the electric field observed proved to be statistically significant and ranged from 0.000 V/m to 3.30 V/m. The mean value in time ranged from a minimum of 0.054 V/m to maximum of 0.101 V/m. The differences described are reflected in the values of the test function (Table 3).

Student’s *t*-test was performed in order to determine statistically significant differences between the values of electric component of the electromagnetic field for the aircraft routes (1,2,3,4) compared to the values obtained for the aircraft located on the ground (background) and is presented in Table 4.

Having compared the values of the electric component of the electromagnetic field for the GSM1800 band, which presented the highest values recorded on different aircraft routes compared to the background (the airport), statistically significant differences were observed.

The next analysis presented included the results obtained during flights with the Cessna C172 aircraft on the route Depułtycze Królewske–Lublin–Krasnystaw–Depułtycze Królewskie for individual frequency ranges. Table 5 presents the characteristics of the electric field intensity E for individual frequency bands of the Cessna C172. A sample group included a total of 7372 measurements registered for each frequency band.

The mean value of the electric field intensity for individual GSM and UMTS frequency bands ranged from 0.011 V/m to 0.074 V/m. The range of the variable analyzed was from 0.000 V/m to 2.18 V/m.

Next, the statistical analysis of the results obtained during flights on AT3 aircraft on the route Depułtycze Królewskie-Lublin-Mielec-Depułtycze Królewskie was performed. The characteristics of the electric field intensity E for individual frequency bands of the AT3 is presented in Table 6. A sample group included a total of 7287 measurements registered for each frequency band.

The mean value of the electric field intensity for individual GSM and UMTS frequency bands ranged from 0.0046 V/m to 0.0629 V/m. The range of the variable analyzed was from 0.000 V/m to 1.17 V/m.

The results obtained during the flight of a Robinsson R44 aircraft were also statistically analyzed. The flight was made on the route Depułtycze Królewskie–Lublin–Kielce–Pyszkowice–Czestochowa–Krasnystaw–Depułtycze Królewskie. A sample group included a total of 14653 measurements registered for each frequency band (Table 7).

The mean value of the electric field intensity for individual GSM and UMTS frequency bands ranged from 0.012 V/m to 0.0687 V/m. The range of the variable analyzed was from 0.000 V/m to 1.89 V/m.

Differences for the indicated frequency bands obtained with the ESM 140 dosimeter proved to be statistically significant and are depicted in Figure 17. Results from flights with different types of aircraft were analyzed for each of the tested frequency ranges. The variations of the electric field intensity for various types of aircraft are shown in Figure 20, Figure 21 and Figure 22.

The data presented in the figures relate to the highest electric field values recorded for four routes made with different types of aircraft in the GSM 900, GSM 1800, and UMTS frequency bands.

## 4. Design of an Onboard EMF Monitoring Sensor System

Although the results obtained confirm that the measured values of the electromagnetic fields in the tested aircraft did not exceed the permissible values, there is a risk related to the high load on the pilot’s body, especially the instructor, who performed a significant number of flights per day. As such, the radiation doses add up and this may cause premature fatigue and other negative effects on the instructor’s mental condition. Bearing in mind the need to increase the level of flight operations safety, it is reasonable to undertake research and development activities to develop a system for monitoring electromagnetic radiation and its impact on pilots during a flight. The authors’ idea is an onboard electromagnetic field monitoring system, shown schematically in Figure 23.

The system uses an electromagnetic radiation sensor and a measurement data recorder built based on a microprocessor system. Communication with the environment is possible via a Bluetooth- and GSM-based radio system. In addition, the system includes an optical pilot fatigue sensor using an iris image of the eye. The whole system functions as part of an Internet of Things, thanks to which, it is possible to remotely read, configure, set alerts, etc. Adjusting the system to the needs of a particular case is also an important feature, taking into account various types of aircraft, pilots’ workload, and other factors.

The system can be installed anywhere in the aircraft cabin, though the radiation sensor should be placed close to the pilot, for example, on the back of the seat.

Developmental works on the presented idea of the electromagnetic field-monitoring system will be the subject of the authors’ continued and future research.

## 5. Conclusions

The results of the measurements were compared to the 2013/35/UE directive described in Michalowska et al. [1]. The maximum values recorded in the bands, which ranged from 2 GHz to 6 GHz (GSM 1800, UMTS) and from 400 MHz to 2 GHz (GSM 900), were within safe limits accepted by the directive, and some of the values obtained were several-fold lower. The ESM 140 dosimeter indicated the maximum value for the GSM 1800 band of E = 3.30 V/m during the flight of the Cessna C152, which was registered during flights with all types of aircraft. The readings were taken when the Cessna C152 aircraft was located above Bilgoraj. The maximum value of the electric component of the electromagnetic field E = 2.0 V/m was recorded using the GSM 900 frequency during the Robinson R44 flight. Values registered with the electric component dosimeter for the UMTS frequency band were for all types of aircraft of similar order. The maximum value of E = 2.59 V/m was also registered during the longest flight of the Robinson R44 helicopter. The dosimeter showed the lowest values for the GSM 900 frequency with E = 1.98 V/m, also for the R44 helicopter.

It is worth highlighting that the values of the electric component intensity of the electromagnetic field recorded close to the ground were considerably lower compared to the readings taken during the flight. Similar findings were confirmed in the course of the statistical analysis. 

Field intensity levels in the vicinity of antenna masts were caused by radio communication of a divergent character. The levels registered depended to a great extent on the power of the antennas within the base stations. Due to the character of the transmission and construction of the antennas, the distribution of the field intensity was directional. Although it is difficult to determine the effect of these levels on the environment and its living organisms, they should be constantly monitored. Most of the results presented refer to the measurement performed on board of Cessna C152 aircrafts. It is worth emphasizing that similar results were found for four similar aircrafts. Quantitative measurements of the electromagnetic field intensity were most frequently conducted on the ground or in its close vicinity at a maximum of several dozen meters. A different methodology was utilized in the study where the electric component of the electromagnetic field was analyzed at height ranging from 0 to 860 m. 

Measurement with the dosimeter did not influence board instruments nor avionic units. Measurement antennas were placed outside this type of aircraft. The ESM 140 dosimeter did not generate any high-frequency radiation and thus is suitable for electro-sensitive persons and epidemiological studies considering a minimum field intensity.

## Figures and Tables

**Figure 1 sensors-19-05537-f001:**
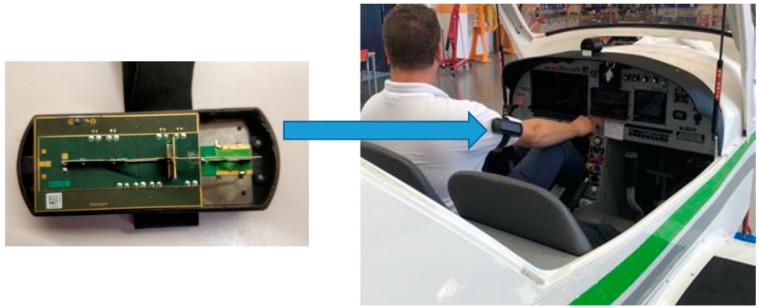
Measuring system for obtaining the intensity of the electromagnetic field from electrical components using an EMS 140 dosimeter during the flight: (**left**) the ESM 140 dosimeter used for the measurements, and (**right**) the location of the device during test flights.

**Figure 2 sensors-19-05537-f002:**
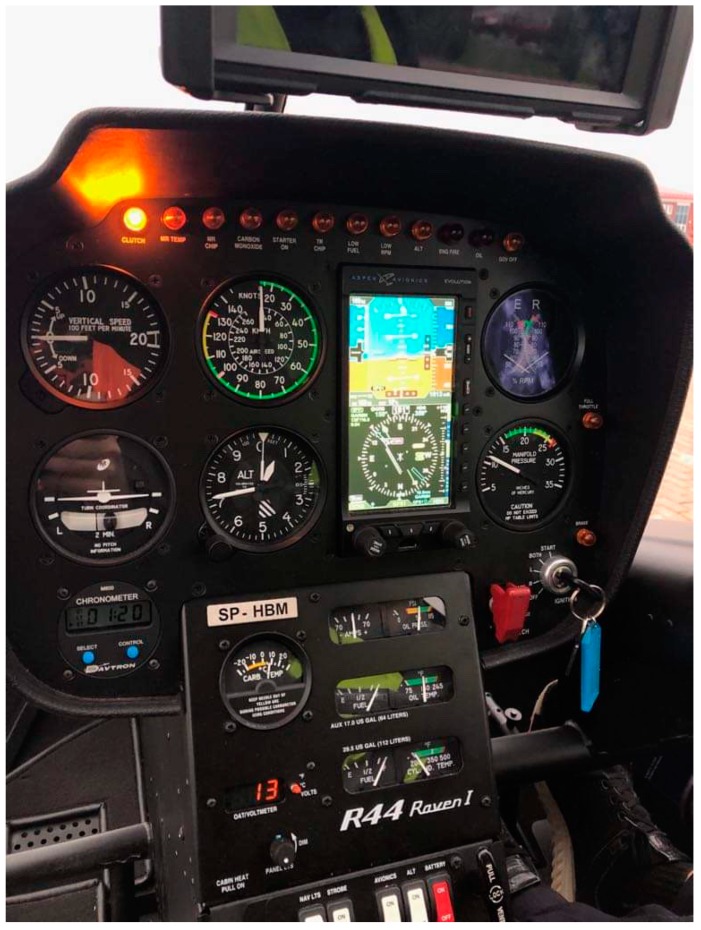
Avionics of the Robinson R44 Raven helicopter.

**Figure 3 sensors-19-05537-f003:**
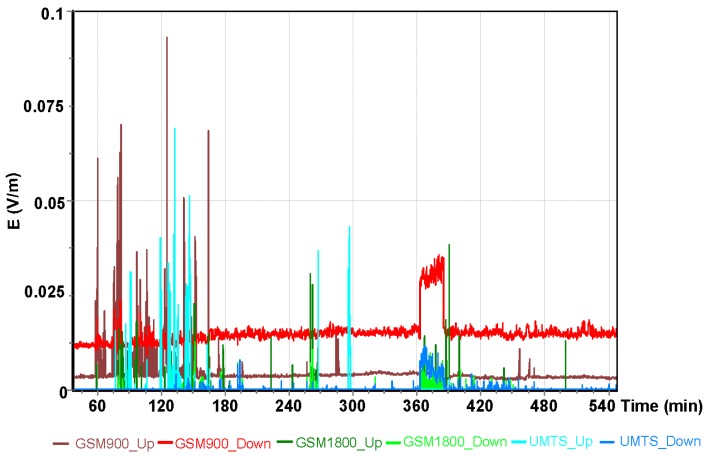
The intensity of the electric field for high frequencies on the Air Traffic Control Tower.

**Figure 4 sensors-19-05537-f004:**
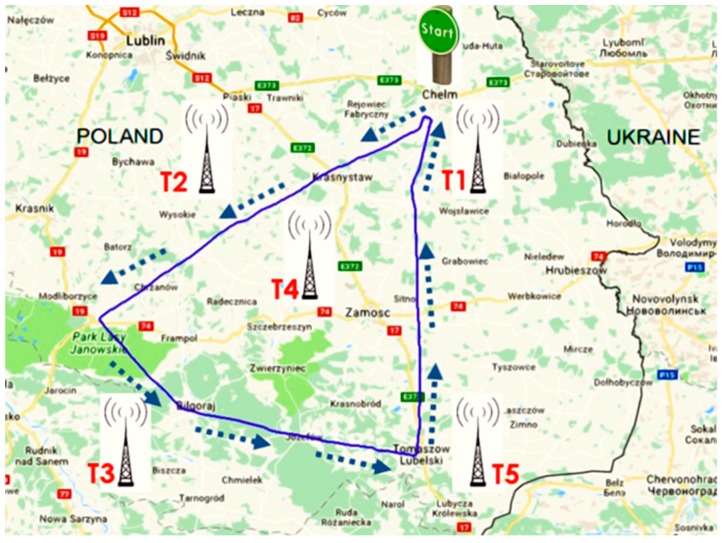
The flight route of the Cessna C152.

**Figure 5 sensors-19-05537-f005:**
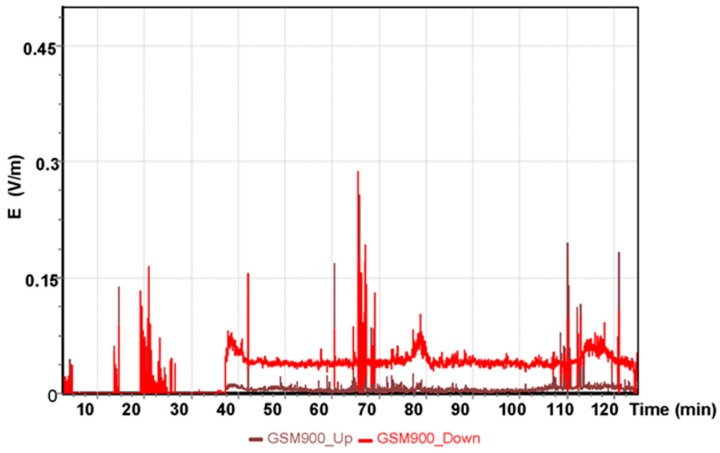
The intensity of the electrical component of the electromagnetic field in the course of a flight for the GSM 900 system.

**Figure 6 sensors-19-05537-f006:**
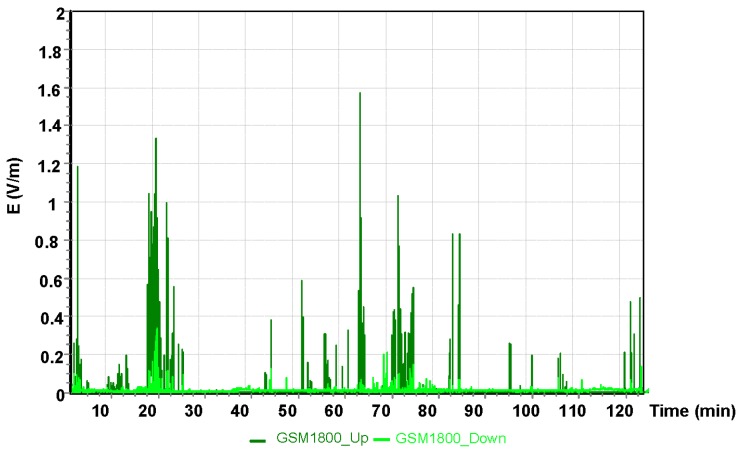
The intensity of the electrical component of the electromagnetic field in the course of a flight for the GSM 1800 system.

**Figure 7 sensors-19-05537-f007:**
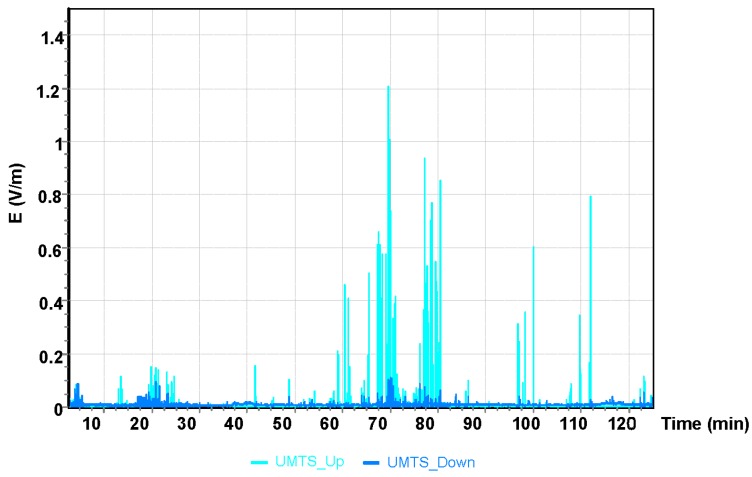
The intensity of the electrical component of the electromagnetic field in the course of a flight for the UMTS frequency.

**Figure 8 sensors-19-05537-f008:**
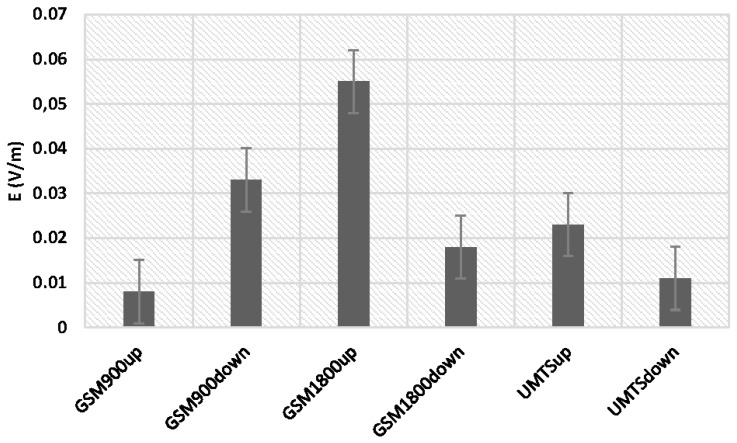
Measurement error of the electrical component of the electromagnetic field intensity during a flight for the frequencies researched.

**Figure 9 sensors-19-05537-f009:**
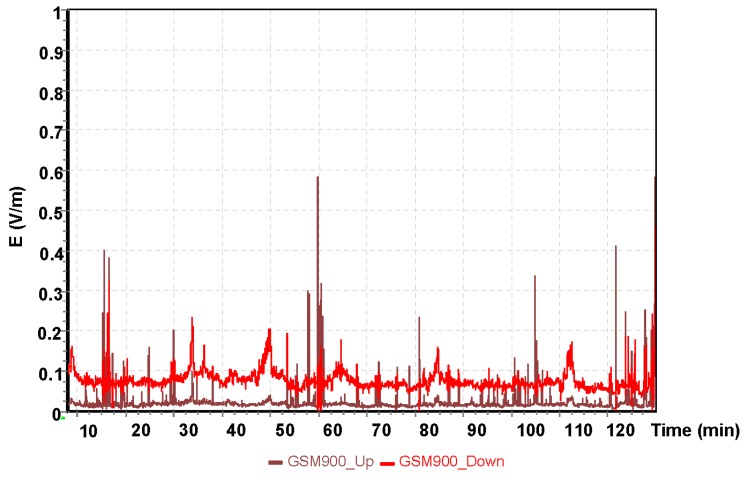
The intensity of the electrical component of the electromagnetic field during a flight of the Cessna C172 for the GSM 900 system.

**Figure 10 sensors-19-05537-f010:**
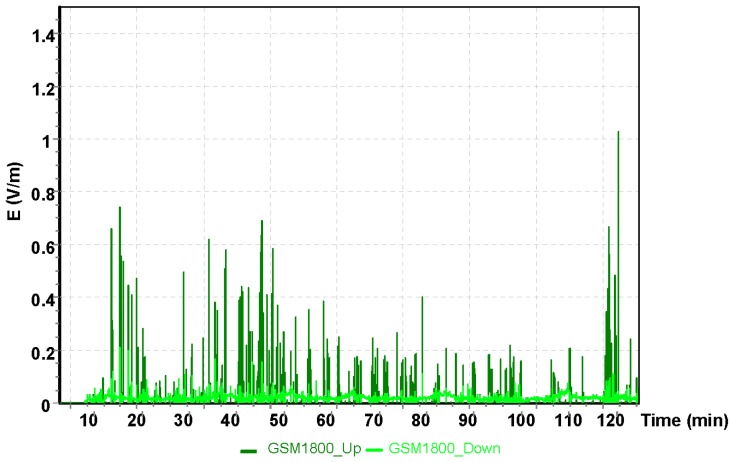
The intensity of the electrical component of the electromagnetic field during a flight of the Cessna C172 for the GSM 1800 system.

**Figure 11 sensors-19-05537-f011:**
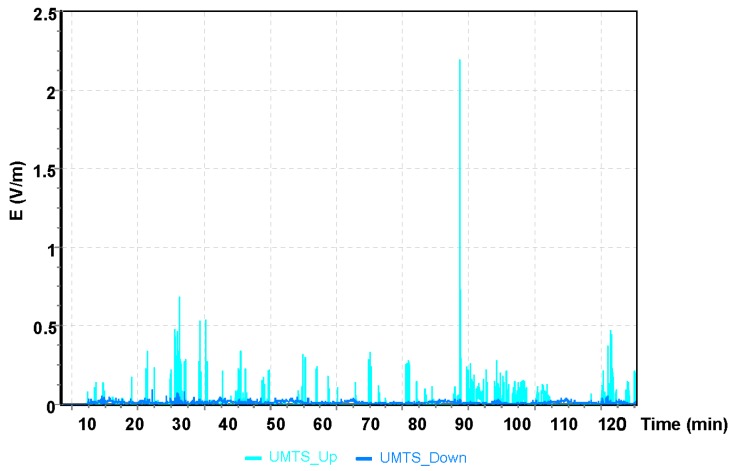
The intensity of the electrical component of the electromagnetic field during a flight of the Cessna C172 for the UMTS frequency.

**Figure 12 sensors-19-05537-f012:**
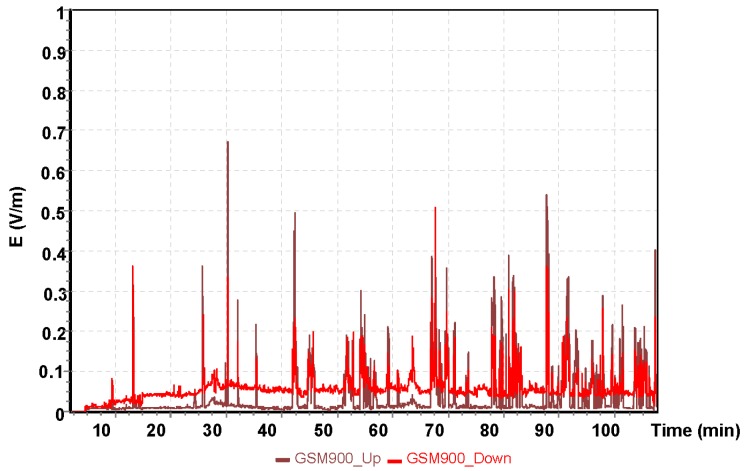
The intensity of the electrical component of the electromagnetic field during a flight of the AT3 for the GSM 900 system.

**Figure 13 sensors-19-05537-f013:**
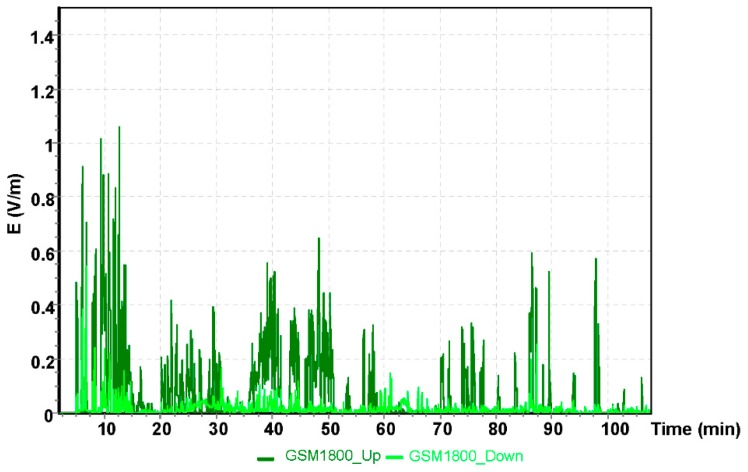
The intensity of the electrical component of the electromagnetic field during a flight of the AT3 for the GSM 1800 system.

**Figure 14 sensors-19-05537-f014:**
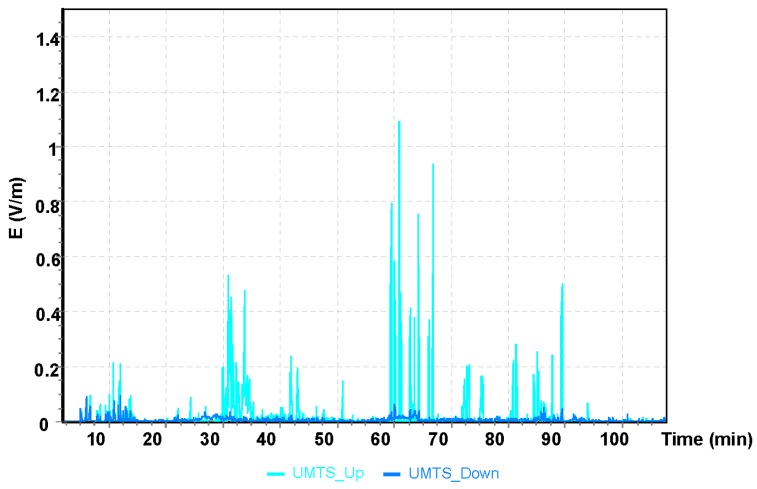
The intensity of the electrical component of the electromagnetic field in the course of a flight of the AT3 for the UMTS frequency.

**Figure 15 sensors-19-05537-f015:**
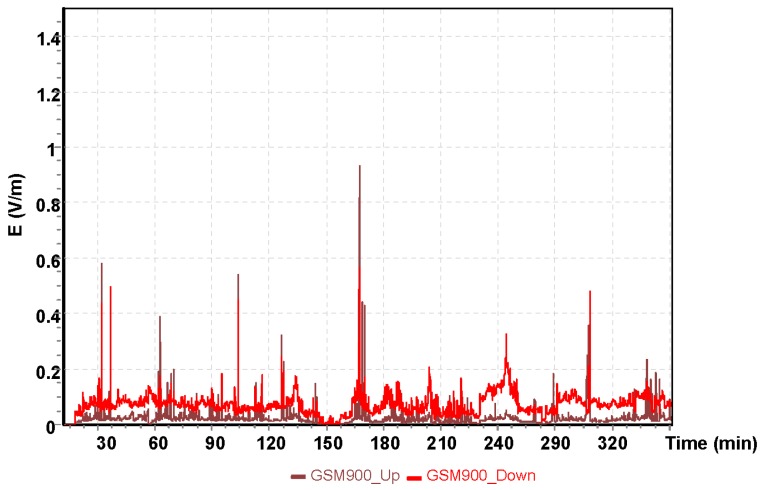
The intensity of the electrical component of the electromagnetic field during a flight of the Robinson R44 for the GSM 900 frequency.

**Figure 16 sensors-19-05537-f016:**
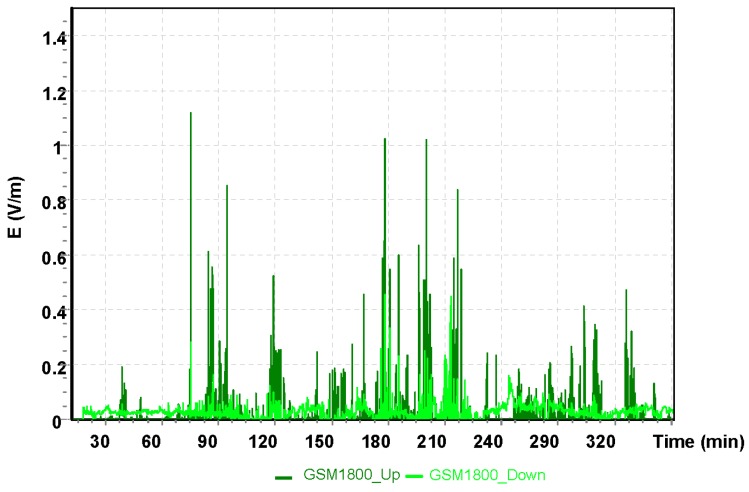
The intensity of the electrical component of the electromagnetic field during a flight of the Robinson R44 for the GSM 1800 frequency

**Figure 17 sensors-19-05537-f017:**
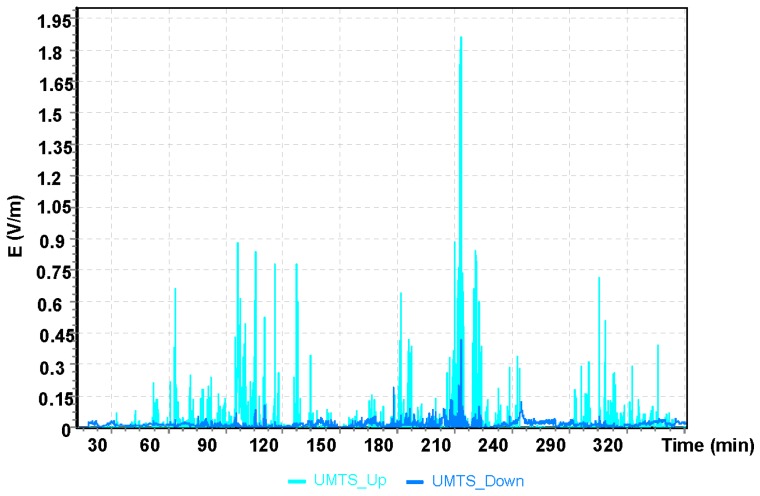
The intensity of the electrical component of the electromagnetic field during a flight of the Robinson R44 for the UMTS frequency.

**Figure 18 sensors-19-05537-f018:**
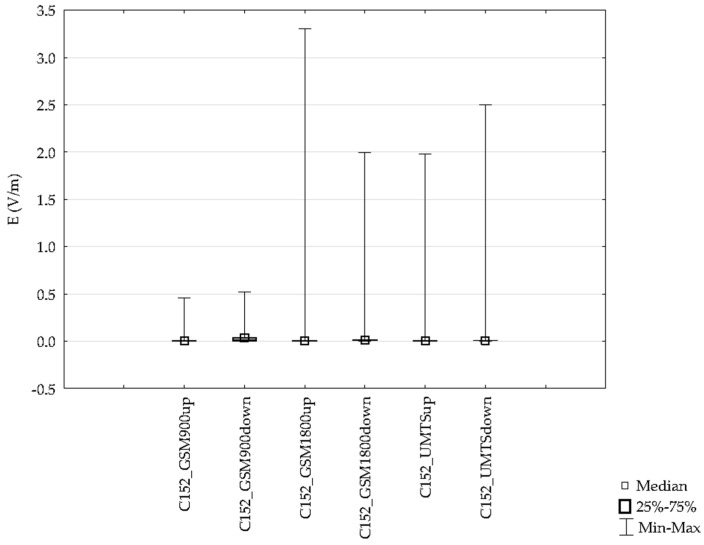
Range of the variability and median the intensity of electric field for high-frequency on the pilot during an airplane flight for one route.

**Figure 19 sensors-19-05537-f019:**
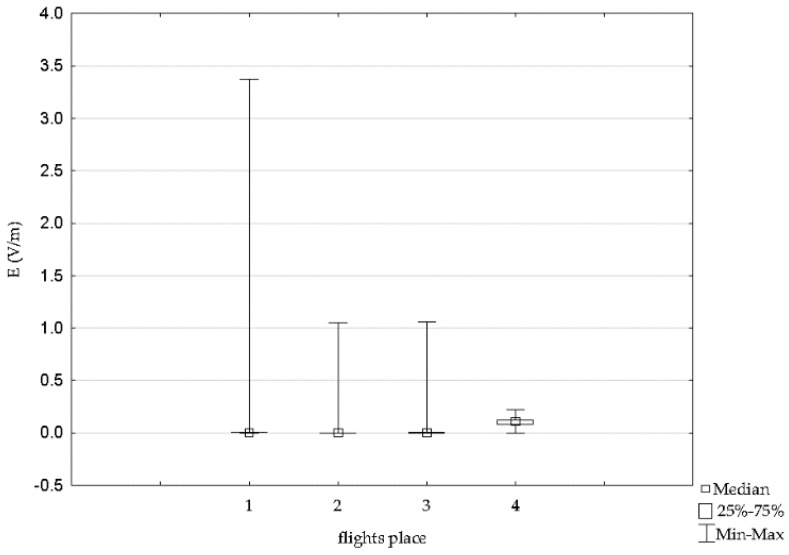
Range of variability and median the intensity of electric field for high-frequency on the pilot during an airplane flight of the Cessna C152 for four routes.

**Figure 20 sensors-19-05537-f020:**
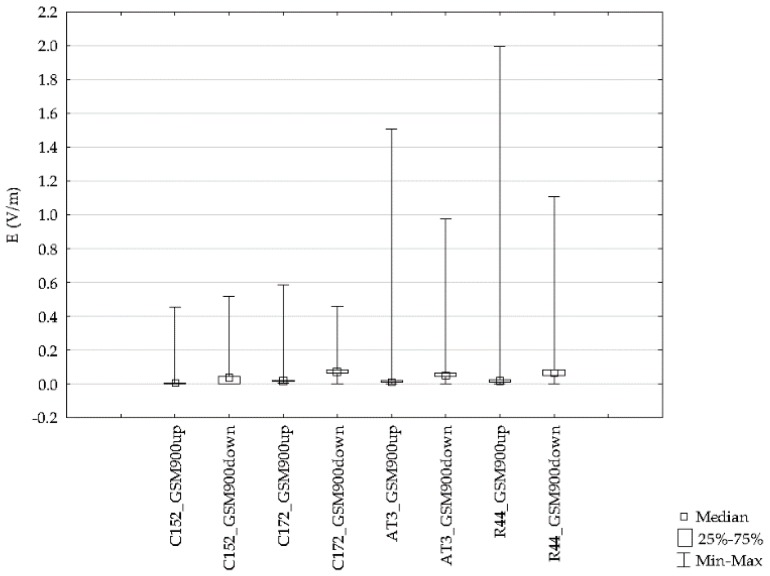
Range of the variability and median during a GSM 900 frequency airplane flight for different routes.

**Figure 21 sensors-19-05537-f021:**
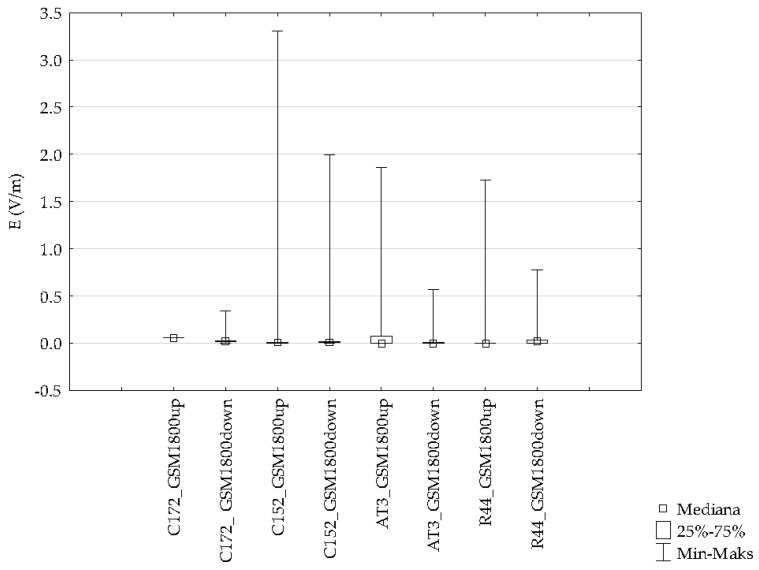
Range of the variability and median during a GSM 1800 frequency airplane flight for different routes.

**Figure 22 sensors-19-05537-f022:**
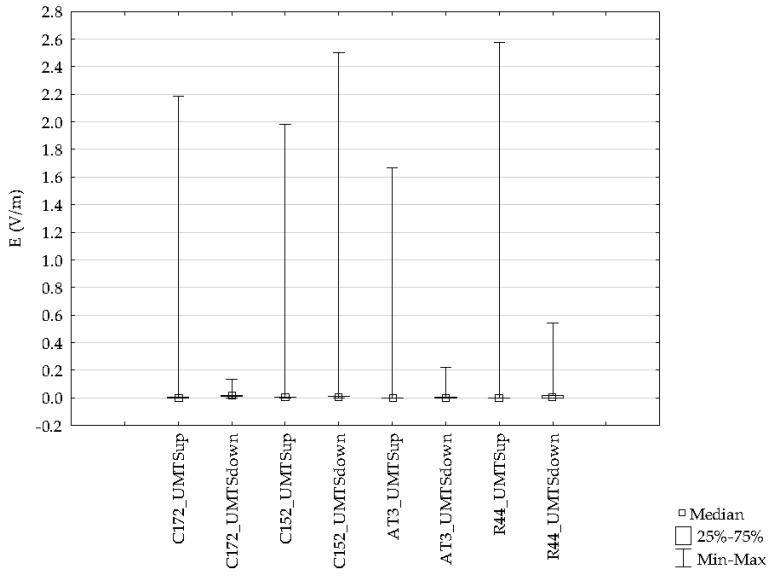
Range of the variability and median during a UMTS frequency airplane flight for different routes.

**Figure 23 sensors-19-05537-f023:**
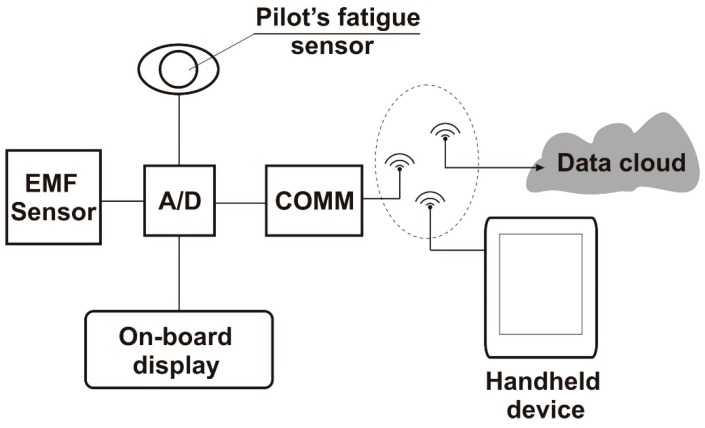
Onboard electromagnetic field-monitoring system.

**Table 1 sensors-19-05537-t001:** Basic technical data of the aircrafts used for the study.

Type	No. of Seats	Wingspan/Rotor Diameter (m)	Engine (kW)	Avionics	Other
**Fixed-Wing Aircrafts**					
Cessna 152	2	10.11	86	Garmin G5	
Cessna 172	4	11.00	125	GarminG1000	
AERO AT3	2	7.55	75	Garmin G500TXI	VLA—Very Light Aircraft Class
**Rotary-Wing Aircraft**					
R44 Raven	4	9.00	183	Mixed: Analog/Aspen electronic flight instrument system (EFIS)	

**Table 2 sensors-19-05537-t002:** Characteristic of the intensity of the electrical component of the electromagnetic field E (V/m) for route 1 with the Cessna C152.

Variable	Mean	Minimum	Maximum	SD
GSM900up	0.0072	0.000	0.4539	0.0194
GSM900down	0.0315	0.000	0.5178	0.0260
GSM1800up	0.0480	0.000	3.3078	0.2080
GSM1800down	0.0174	0.000	0.9989	0.0595
UMTSup	0.0158	0.000	1.9821	0.0736
UMTSdown	0.0103	0.000	2.5000	0.0340

**Table 3 sensors-19-05537-t003:** The characteristics of the electric field intensity E (V/m) for the routes 1, 2, 3, and 4.

Variable	Mean	Minimum	Maximum	SD
1	0.0540	0.0002	3.3078	0.2675
2	0.0142	0.0000	1.0467	0.0670
2	0.0192	0.0000	1.0593	0.0568
4	0.1019	0.0002	0.2174	0.0374

**Table 4 sensors-19-05537-t004:** Test results for individual comparisons (GSM1800).

Test Results for Individual Comparisons		Statistical Significance
Background vs. 1	*p* = 0.000	Relevant
Background vs. 2	*p* = 0.000	Relevant
Background vs. 3	*p* = 0.000	Relevant
Background vs. 4	*p* = 0.000	Relevant

**Table 5 sensors-19-05537-t005:** Characteristics of the intensity of the electrical component of the electromagnetic field E (V/m) with the Cessna C172.

Variable	Mean	Minimum	Maximum	SD
GSM900up	0.0229	0.0031	0.5821	0.0311
GSM900down	0.0749	0.0000	0.4582	0.0246
GSM1800up	0.0602	0.0602	1.1602	0.0000
GSM1800down	0.0197	0.0000	0.3463	0.0133
UMTSup	0.0184	0.0000	1.8870	0.0704
UMTSdown	0.0114	0.0000	0.1360	0.0074

**Table 6 sensors-19-05537-t006:** Characteristics of the intensity of the electrical component of the electromagnetic field E (V/m) for a flight with the AERO AT3 aircraft.

Variable	Mean	Minimum	Maximum	SD
GSM900up	0.0335	0.0000	0.6838	0.0311
GSM900down	0.0612	0.0000	0.5155	0.0246
GSM1800up	0.0629	0.0000	1.1507	0.0000
GSM1800down	0.0098	0.0000	0.5669	0.0133
UMTSup	0.0195	0.0000	1.1697	0.0704
UMTSdown	0.0046	0.0000	0.2169	0.0074

**Table 7 sensors-19-05537-t007:** Characteristics of the intensity of the electrical component of the electromagnetic field E (V/m) with the Robinsson R44.

Variable	Mean	Minimum	Maximum	SD
GSM900up	0.0210	0.0000	0.9839	0.0357
GSM900down	0.0687	0.0000	0.5182	0.0401
GSM1800up	0.0205	0.0000	1.1545	0.0861
GSM1800down	0.0235	0.0000	0.5748	0.0313
UMTSup	0.0230	0.0000	1.8976	0.1063
UMTSdown	0.0127	0.0000	0.4411	0.0186

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
