# Peer review of "Monitoring the Risk of the Electric Component Imposed on a Pilot During Light Aircraft Operations in a High-Frequency Electromagnetic Field"

_sensors, 2019, doi:10.3390/s19245537_

Round 1

Reviewer 1 Report

Reviewer’s report

General assessment: This paper presents the results of measurements and statistical analysis assessment of electric intensity performed during flights of four ultra-light, sport aircrafts Cessna C152, Cessna C172, Aero AT3 R100, and the Robinson R44 Raven helicopter. The conclusion has been drawn that the maximum values of the electric intensity are within safe limits.

Technical comments:

Page 1, line 23: It is not necessary to use the article “a” in the following listing of the aircrafts: “a Cessna C152, a Cessna C172, an Aero AT3 R100 and a Robinson”. Please remove the indefinite article “a” in this list. Page 1, line 39: please provide noun-verb correspondence by changing “are” to “is” in the following sentence: “Moreover, research on the effect of electromagnetic field on human health, which are 40 concerned with the operation of electric and electronic devices, are being conducted [3-5].” Page 2, line 68-69: Please notice that avionics is a singular noun to be used in the sentence: “Integrated avionics, which uses one large display instead of many single indicators, is becoming increasingly popular.” Page 2, line 74: Change preposition “from” to “of” in the following sentence: “The increasing load from the electromagnetic field …” Page 2, line 83: Please define in the course of what the flights mapping was performed. I guess it should state “in the course of this work, the flights mapping…” Page 2, line 93: Please explain the abbreviation “PEM”. Page 4, line 136: Delete “there was” in the sentence: “The entire work space has been subjected to the measurement including all areas where there was the electromagnetic fields could be present.” Page 4, line 148: Use article “the” instead of “a” in the sentence started with “A Cessna 152 and Cessna 172 aircrafts have Glasscocpit avionics, …” Page 6, the caption of Figure 2: Please add AERO AT3 to the list of aircrafts. Page 4, line 206: Add “to the” to the sentence ending: “were regarded to the GSM 900 frequency band.” Excessive use of the article “a” in phrases on Page 8, line 216 (“the dosimeter in a Cessna 152.”) and Fig. 6 caption “in the course of a flight for the GSM 900 system.” – in these cases article “a” could be deleted. The absence of Figure 14 disturbs the analysis of the highest and lowest electrical fields for GSM 900 and 1800. Please renumber the figures and correct the corresponding analysis of the electric fields presented on page 13. Please correct the caption of Figure 21. I guess it should be written something like the following: “Range of variability and median during high-frequency airplane flight of the Cessna C152 for four routes”. Page 16, line 528: Please correct the beginning of the sentence: “Student’s t- test was performed…” Ending of this sentence with several brackets could be improved as well, adding a verb will make it easier for reading, I propose to end it with a phrase “as it is presented in Table 6.” Inconsistency of Figure 18 and its caption has been found on page 17, line 567, because Fig. 18 depicted other values. Please add the correct figure or remove this sentence.

Author Response

Authors’ reply to Reviewers’ comments

Manuscript: Sensors 657854

Title: Monitoring the Risk of Electric Component Imposed on a Pilot During the Light Aircraft Operation in High Frequency Electromagnetic Field  

Authors: Joanna Michałowska, Arkadiusz Tofil, Jerzy Józwik, Jarosław Pytka, Stanisław Legutko, Zbigniew Siemiątkowski and Andrzej Łukaszewicz

Reviewer #1

Thank you for your comments. We’ve corrected the text in all places you’ve indicated and according to your suggestions.

Thank you once again for your time and effort to evaluate our manuscript.

Sincerely,

Authors

Reviewer 2 Report

First of all, congratulations for this excellent achievement and interesting results.

I think this manuscript has scientific and engineering merits to be accepted by this journal.
I would suggest some revisions.
Please consider to implement such modifications, if applicable.

[General Comments]
It looks some parts are not significantly necessary and just redundant.
For example 1), Lie 59-80 is not related to the main subject of this manuscript.
Of course, as a general explanation, this is useful.
It, however, is not necessary to contain full of this description, just a brief summary is enough.
For example 2), Line 97-107. Is it really necessary to discuss this manuscript ?
Your manuscript is not a textbook of electromagnetism.
For example 3)l Secion 2.3.
I would suggest that this subsection can be merged into Section 2.2 by reducing a volume of description.
Again, your manuscript is not a catalogue of aircrafts.
By reducing such redundant descriptions, readability of this manuscript would be much better than present.

At next, although the consideration on statistical treatment is given in detail, discussion on the measurement accuracy looks not so much. What is the source of measurement uncertainty listed in Figure 9.
Without such discussion, this manuscript provides only the measurement result.
Of course, your measurement result is fruitful by itself.
However, this journal is “Sensors”, only the measurement result is not fruitful for readers.

[Technical Comments]
Line 92, “PEM” needs full description, not good use abbreviation at the first appearance.
Line 109, if possible, it is better to refer some article to get more info about ESM140 here.

Author Response

Authors’ reply to Reviewers’ comments

Manuscript: Sensors 657854

Title: Monitoring the Risk of Electric Component Imposed on a Pilot During the Light Aircraft Operation in High Frequency Electromagnetic Field  

Authors: Joanna Michałowska, Arkadiusz Tofil, Jerzy Józwik, Jarosław Pytka, Stanisław Legutko, Zbigniew Siemiątkowski and Andrzej Łukaszewicz

Reviewer #2

Thank you for your comments. Our reply is included below and the requested corrections have been included in the revised version of the manuscript.

The text within the lines 58 – 80 (original manuscript) has been rewritten and the new version is as follows:

The increasing load of the electromagnetic field in light aircrafts can lead to negative effects on pilots' health and mental condition, especially when it comes to instructors flying many-hour flights daily, which can lead to safety risks. Studies on the impact of electromagnetic fields on humans result from the EU Directive 2013/35/ directive. The continuous increase in radio infrastructure, including mobile telephony, is associated with an increasing electromagnetic field strength. Especially in the area around GSM and UMTS relays, the electromagnetic field values ​​can reach high values. The second source of electromagnetic field that affects the pilot during the flight is avionics. Integrated avionics, which use one large display instead of many single indicators, is a great convenience for the pilot, due to the presentation of many flight  information on one display. Glasscocpit is currently the standard in aircraft communication. Thus, Flight Training Organizations (FTO) are increasingly willing to use training aircrafts with the Glasscocpit system.

The purpose of the present study was to conduct electromagnetic field measurements on selected aircrafts of the Aviation Training Center in Royal Depultycze, near Chelm, Poland. The center is an integral part of the higher vocational school in Chelm and conducts training for airline pilots on airplanes and helicopters as a part of engineering studies.

Similarly, the text in the lines 97 – 107 (original manuscript) has also been fully changed into:

The Maschek ESM 140 dosimeter was used for the flight test measurements. The device measures the electromagnetic field for broadband high frequencies in real time and the measured data was stored in the device’s memory. We’ve transferred the data to a personal computer for analysis after each flight. The measuring range was 0.01 to 70 V/m, sensitivity of 10 mV/m and accuracy of ± 2 dB in free field and 4dB, when the device was installed on the pilot’s arm for in-flight measurements (see Figure 1 right).

We’ve deleted the fotographs showing the aircrafts used in the study.

We’ve added a text that clarifies the matter of measuring error (lines 240 – 248):

The limits of measurement uncertainty for the data obtained are given in Figure 8. Systematic error was chosen as the basis for uncertainty, whose sources are mainly: instrument class, characteristics of band filters in the tested frequencies, flight altitude and heading relative to electromagnetic field sources as well as meteorological factors. Moreover, the measurement error was also affected by the installation of the measuring unit, which, mainly for practical reasons, was placed on the pilot's arm (see Figure 1 right), which meant that the measurement was not carried out in the free field. In future test, we’re planning to use an on-board electromagnetic field monitoring sensor in order to minimize the installation effects. This sensor is described in the Section 5 of the present paper.

Thank you once again for your time and effort to evaluate our manuscript.

Sincerely,

Authors
